# Swarm Intelligence Enhanced Reasoning: A Density-Driven Framework for LLM-Based Multi-Agent Optimization

## Abstract

Recently, many approaches, such as Chain-of-Thought (CoT) prompting and Multi-Agent Debate (MAD), have been proposed to further enrich Large Language Models' (LLMs) complex problem-solving capacities in reasoning scenarios. However, these methods may fail to solve complex problems due to the lack of ability to find optimal solutions. Swarm Intelligence has been serving as a powerful tool for finding optima in the field of traditional optimization problems. To this end, we propose integrating swarm intelligence into the reasoning process by introducing a novel Agent-based Swarm Intelligence (ASI) paradigm. In this paradigm, we formulate LLM reasoning as an optimization problem and use a swarm intelligence scheme to guide a group of LLM-based agents in collaboratively searching for optimal solutions. To avoid swarm intelligence getting trapped in local optima, we further develop a Swarm Intelligence Enhancing Reasoning (SIER) framework, which develops a density-driven strategy to enhance the reasoning ability. To be specific, we propose to perform kernel density estimation and non-dominated sorting to optimize both solution quality and diversity simultaneously. In this case, SIER efficiently enhances solution space exploration through expanding the diversity of the reasoning path. Besides, a step-level quality evaluation is used to help agents improve solution quality by correcting low-quality intermediate steps. Then, we use quality thresholds to dynamically control the termination of exploration and the selection of candidate steps, enabling a more flexible and efficient reasoning process. Extensive experiments are conducted on widely-used seven mathematical reasoning benchmarks, i.e., MATH-500, MMLU-STEM, etc. As expected, our method consistently outperforms both CoT methods and existing reward-guided approaches, particularly on complex problems. This demonstrates the effectiveness of our approach in leveraging swarm intelligence for enhanced reasoning.

## 1 Introduction

Large Language Models (LLMs) have demonstrated remarkable progress across various domains, yet significant challenges remain in complex reasoning tasks. Generally, many approaches propose to address the remaining challenges by employing Chain-of-Thought (CoT) prompting Wei et al. (2022a); Kojima et al. (2022) or its variants Wang et al.; Yao et al. (2023); Zhou et al.; Zhang et al.; Wang et al. (2023), guiding models to decompose complex problems into manageable steps, and reasoning the final answer step by step. However, Large language models are typically prone to making hallucinations during the CoT reasoning process due to inherent randomness and limitations in comprehension, which may lead to incorrect results.

Alternatively, researchers propose to use Multi-Agent Debate (MAD) frameworks to perform interactions among multiple agents, collaboratively deriving better answers Du et al. (2023); Liang et al. (2024); Chen et al. (2024); Wang et al. (2024). Generally, a variety of agents are defined and assigned distinct, pre-defined roles. They then follow a designated communication protocol or scheme to collaboratively generate better answers. Note that the success of MAD is presumably based on the hypothesis that LLMs can take on heterogeneous roles with various human-crafted prompts. Different from CoT methods, the heterogeneous prompts can help agents think divergently together, expanding the extent of exploration of the solution space and ultimately converging on

a better solution. However, this hypothesis is not always valid and frequently fails, making MAD ineffective in deriving better answers. Instead, MAD methods even perform worse than basic CoT methods. As mentioned in Zhang et al. (2025a), the factor may be attributed to the fact that various agents are inherently homogeneous. Despite the different roles of the agents, they still rely on the same large language model. Solutions generated by different agents hold high similarities. Moreover, during the collaboration process, influenced by the context constructed from historical interaction information, the similarity between the solutions generated by the agents gradually increases, which is significantly lower than the diversity of using multiple CoTs with the same computational resource consumption (the details are shown in Appendix C), thus making it difficult to find the optimal result. This leads us to a simple but crucial question: how can we generate more diverse solutions to increase the chances of finding the correct answer?

Swarm intelligence algorithms are a kind of traditional heuristic optimization method Bonabeau et al. (1999); Kennedy & Eberhart (1995); Dorigo et al. (2007) that mimic the collective behaviors observed in nature. Generally, they begin by forming an initial population of candidate solutions through repeated sampling in the solution space, and then gradually improve the population quality using feedback from fitness value indicators. With limited computational resources, they show significant effectiveness compared to direct random sampling. Inspired by this, we conceptualize LLM reasoning as solution space exploration, treating the reasoning process as an optimization problem where a population of LLM-based agents collectively searches the solution space for optimal solutions.

However, directly applying swarm intelligence algorithms may result in suboptimal solutions. Due to the complexity of LLM reasoning and the high diversity of possible solutions, the solution space contains multiple global optima with correct reasoning pathways alongside local optima with incorrect reasoning pathways. Traditional swarm intelligence algorithms tend to cause population concentration within the same region of the solution space, resulting in convergence to the same optima and diminishing the ability to thoroughly explore the space as population diversity decreases. Consequently, once trapped in an incorrect local optimum, the population loses its capability to discover global optimal solutions. Despite existing research Wei et al. (2022b); Ahrari et al. (2022); Zhu et al. focused on maintaining population diversity, it remains crucial to be aware of the risk of local optima entrapment when applying swarm intelligence algorithms to LLM reasoning tasks.

In this paper, we propose a novel paradigm called Agent-based Swarm Intelligence (ASI), which conceptualizes LLM reasoning as a solution search process in the solution space for global optima. ASI consists of a generator $G$ generating solutions and an evaluator $E$ evaluating the step-level quality of solutions. Based upon ASI, we address the aforementioned local optima issue by introducing the density-assisted search mechanism and developing the Swarm Intelligence Enhancing Reasoning (SIER) framework. This mechanism employs kernel density estimation techniques and non-dominated sorting to implement step-level Pareto front selection that jointly optimizes steps' quality and density, enabling exploring solution space efficiently while preserving diversity. Through this innovative mechanism, SIER efficiently enhances solution space exploration by expanding the diversity of the search paths while mitigating convergence to local optima. Furthermore, a step-level quality evaluation is used to enable agents to enhance solution quality by rectifying intermediate steps of low quality. Then, we use the termination of exploration and the sampling of candidate steps can be dynamically controlled through quality thresholds, facilitating a more flexible and efficient reasoning process. We conduct extensive experiments on seven widely used mathematical reasoning benchmarks, including AIME-2024, AIME-2025, MATH-500, and other challenging datasets. Across various evaluation metrics, SIER consistently outperforms state-of-the-art methods, demonstrating the effectiveness of our approach in enhancing reasoning capabilities.

## 2 RELATED WORK

### 2.1 MULTI-AGENT SYSTEMS

Recently, LLM-based Multi-Agent SystemsGuo et al. (2024); Tran et al. (2025); Han et al. (2024) have rapidly emerged as powerful frameworks for solving complex problems. Early developments like Camel Li et al. (2023) and MetaGPT Hong et al. provided infrastructure for agent collaboration. Later on, the Multi-Agent Debate (MAD) strategy Wu et al.; Chen et al.; Zhou et al. (2025) has been widely adopted to facilitate collaborative interactions among agents through discussion. Several representative approaches include: Society of Mind (SoM) Du et al. (2023) establishes a three-step framework for agents to debate and reconcile differences, enhancing factuality through consensus

formation. Multi-Persona Liang et al. (2024) introduces contrasting "angel" and "devil" roles, encouraging diverse perspectives and creative problem-solving, while Exchange-of-Thoughts (EoT) Yin et al. (2023) are designed to facilitate cross-communication between models to enhance collective understanding of the problem-solving process. In addition, COMM Chen et al. (2024) adopts different reasoning paths for different roles to implement few-shot prompting approaches in multi-agent scenarios, effectively enhancing the performance in domain-specific tasks.

While the MAD scheme has shown effectiveness, previous systematic evaluations Zhang et al. (2025a) reveal that many MAD methods perform even worse than CoT methods. The advantage of solution diversity introduced by heterogeneous agent roles does not always hold, as the agents ultimately rely on the same underlying large language model, which leads to high similarities of the solutions generated by agents. Furthermore, the similarity between the solutions gradually increases during the collaboration process, since the LLM is progressively influenced by the accumulated context from historical interaction information. This factor makes it difficult to find the optimal results in the final despite the use of a collaborative scheme. Different from them, we propose integrating swarm intelligence algorithms into the reasoning process to enhance the discovery of optimal solutions. We conceptualize the LLM's reasoning process as an exploration of the solution space and adopt heuristic algorithms to efficiently identify optimal solutions.

## 2.2 Swarm Intelligence Algorithms

Swarm intelligence algorithms Storn & Price (1997); Holland (1992) are biologically inspired optimization methods that solve complex problems through natural selection and genetic variation. The evolutionary scheme involves the generation of initial candidate solutions, fitness evaluation, adaptive selection, and the creation of new candidates.

Although swarm intelligence algorithms can effectively enhance solution space exploration capabilities, they sometimes cause populations to converge within the same region of the solution space. This factor not only wastes a large amount of computational resources but also increases the risk of getting stuck in local optima, thereby reducing the model's ability to thoroughly explore the solution space. To address this problem, existing work Wei et al. (2022b); Ahrari et al. (2022); Zhu et al.; Luo et al. (2022); Xu et al. (2021) has tried various strategies to maintain population diversity. However, this may not be applicable to LLM reasoning because the agent itself has the ability to generate solutions using LLM step-by-step thinking, and many of these strategies cannot directly affect the solution generation. Inspired by traditional swarm intelligence algorithms, we propose a novel paradigm called ASI and develop the SIER framework, which evaluates reasoning steps based on quality and density metrics, generating multiple high-quality, diverse reasoning paths that enhance the solution space exploration for complex problems, thereby improving problem-solving capabilities.

## 3 Preliminary

### 3.1 Kernel Density Estimation

Kernel Density Estimation (KDE) is a fundamental non-parametric technique for probability density estimation from finite samples Wand & Jones (1994). Given $N$ independent and identically distributed samples $\{\mathbf{x}_i | i = 1, \dots, N\}$, the KDE-based density estimate at a query point $\mathbf{x}$ is given by:

$$\rho(\mathbf{x}) = \frac{1}{N} \sum_{i=1}^{N} K_h(\mathbf{x} - \mathbf{x}_i), \tag{1}$$

$$\text{where } K_h(\mathbf{x} - \mathbf{x}_i) = \begin{cases} \exp\left(-\frac{\|\mathbf{x} - \mathbf{x}_i\|^2}{2h^2}\right), & \text{if } \|\mathbf{x} - \mathbf{x}_i\| \le h, \\ 0, & \text{otherwise.} \end{cases} \tag{2}$$

Here $K_h(\cdot)$ is a smoothing kernel with bandwidth $h$, controlling the locality of the estimation. A common choice is the Gaussian kernel thanks to its desirable properties, e.g., infinite differentiability and exponential decay. Following the widely adopted approach in Xu et al. (2021), we employ a truncated Gaussian kernel. KDE can help our framework construct token-level density landscapes of solution populations, where fully explored areas will have higher density values, and under-explored areas will have lower density values, enabling the identification of over-explored (high-density) and under-explored (low-density) regions in the solution space. It is worth noting that the density information obtained through the Gaussian kernel function in our algorithm does not directly

participate in numerical calculations, but is only used for comparison in the sampling step (we only need to use it to select steps with relatively low density in the sampling reasoning step), so it is relatively insensitive to the choice of kernel function.

## 3.2 MULTI-OBJECTIVE OPTIMIZATION AND NON-DOMINATED SORTING

Multi-objective optimization addresses problems with multiple objectives by identifying a set of Pareto-optimal solutions, collectively forming the Pareto front (Figure 1a). A solution $\mathbf{x}$ is denoted to dominate another solution $\mathbf{x}'$ (denoted $\mathbf{x} \succ \mathbf{x}'$) if it is superior or equal in all $m$ objectives ($\phi_i(\mathbf{x}) \geq \phi_i(\mathbf{x}')$ for $i \in \{1, \ldots, m\}$) and strictly superior in at least one objective ($\phi_j(\mathbf{x}) > \phi_j(\mathbf{x}')$ for some $j$). The Pareto front consists of non-dominated solutions that form a boundary in the objective space. (The details are given in Appendix D.1)

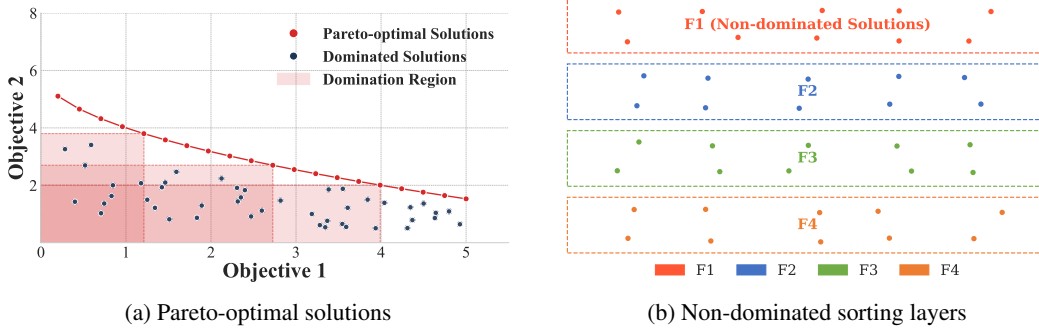

(a) Pareto-optimal solutions    (b) Non-dominated sorting layers

Figure 1: Visualization of Pareto-optimal solutions and non-dominated sorting

Fast non-dominated sorting is an algorithm that categorizes solutions into hierarchical fronts based on these dominance relationships (Figure 1b). The first front, $F_1$, comprises all non-dominated solutions. Subsequent fronts ($F_2, F_3, \ldots$) contain solutions dominated by those in preceding fronts. This sorting technique is a cornerstone of algorithms like NSGA-II Deb et al. (2002).

In our framework, non-dominated sorting is employed to guide the selection of candidate reasoning steps. This process considers two complementary optimization objectives: quality and diversity. Quality refers to the correctness and effectiveness of a reasoning step, while diversity measures its distinctness relative to other reasoning steps. Specifically, diversity is quantified using a density metric, and quality is evaluated by the evaluator $E$. The non-dominated sorting algorithm ranks candidates based on both their quality and this density measure. The selection mechanism subsequently prioritizes candidates on the Pareto front, ensuring that no chosen step is simultaneously outperformed by an alternative candidate with respect to both quality and diversity.

## 4 METHODOLOGY

Swarm intelligence algorithms typically explore the solution space via individual exploration and collective cooperation to identify the global optimum. This objective is analogous to that of LLM-based agents when undertaking reasoning tasks. Consequently, we propos a novel paradigm named ASI, which conceptualized the LLM's reasoning process as an individual's search for the global optimum within a solution space. Specifically, for a given problem $Q$, it owns a solution space $S$. Each dimension corresponds to a token, with its value derived from the LLM's vocabulary. Every point within this solution space denotes a solution in the form of a reasoning path, and the essence of LLM reasoning is to generate such a path. Therefore, we apply swarm intelligence algorithms to LLM reasoning, which leverages the evaluator $E$ to evaluate the quality of the reasoning paths and guide the search, iteratively refining reasoning paths to efficiently and effectively uncover the global optima. Furthermore, based on the ASI paradigm, we propose the Swarm Intelligence Enhancing Reasoning (SIER) framework, which enhances solution space exploration by expanding the diversity of the search paths. This section first presents an overview of the SIER framework and detailed explanations of its key processes.

### 4.1 SWARM INTELLIGENCE ENHANCING REASONING FRAMEWORK

In the SIER framework, LLM-based agents progressively generate reasoning paths that continuously expand the solution search space. This space is strategically navigated through two key mechanisms:

1) the construction of density landscapes, which employs kernel density estimation to guide the agent collective in exploring low-density regions that haven't been fully explored; and 2) a multi-criteria selection mechanism that integrates step-level quality evaluation with density calculations to optimize task performance while maintaining solution diversity.

Specifically, the framework encompasses three main processes: 1) **Population Initialization**: LLM-based agents are created to generate initial reasoning paths and form an initial population; 2) **Population Evolution**: A density landscape is constructed via kernel density estimation. Then, the multi-criteria approach is employed to combine step-level quality evaluation with density calculations to generate high-quality and diverse reasoning paths progressively. 3) **Population Clustering and Selection**: We then cluster the final population based on the answer labels of the reasoning paths, followed by sorting the clusters based on the highest quality of the individuals in the cluster and selecting the $k$ best reasoning paths from each of the first $k$ clusters.

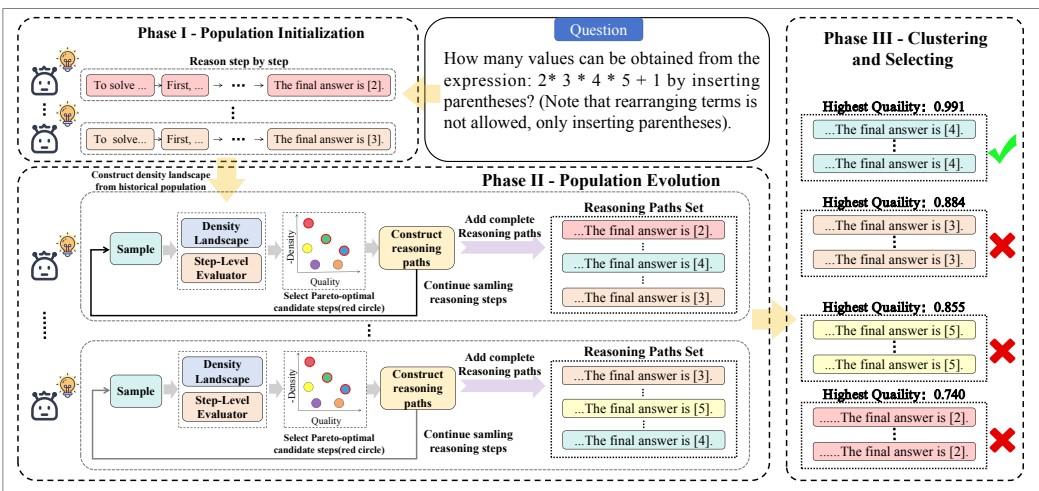

Figure 2: Overview of the SIER framework.

## 4.2 POPULATION INITIALIZATION

Given a task query $q$, our SIER framework first generates the initial population consisting of $n$ individuals. Specifically, $n$ LLM-based agents are created as individuals. Each agent independently processes the input $q$ to generate a distinct step-by-step reasoning path. These reasoning paths collectively form the population's initial solution set. Then, a specialized evaluator (e.g., a Process Reward Model, PRM) will subsequently be used to evaluate each path's quality,

## 4.3 POPULATION EVOLUTION

The population evolution phase involves up to $\mathcal{I}_{max}$ iterations of density-assisted search, generating a new population per iteration. An early stopping criterion terminates this phase if the highest quality observed in the historical population surpasses a threshold $\theta$. This is done because we then believe a sufficiently high-quality solution has been found. If the initial population already meets this criterion, this evolutionary phase is skipped entirely.

**Density-Assisted Search**: We view the reasoning paths generation process as a step-by-step search problem in the solution space. In this space, each step represents a partial reasoning path, starting from an initial empty step and exploring the solution space by adding reasoning steps until a complete reasoning path is found. We employ a multi-path parallel search strategy, where "active paths" represent branches of the solution space currently being explored, with each path starting from an empty step and continuously expanding to explore different regions of the solution space.

To efficiently explore the solution space, we introduce an adaptive sampling mechanism. For each active path, we first use the LLM to incrementally sample a small set of candidate expansion steps and compute their quality using the evaluator $E$. If the quality score of a candidate step exceeds a threshold, we immediately select that step and prune the other branches; otherwise, we continue to expand the search space by resampling more candidate steps. This approach dynamically adjusts the

search breadth according to the complexity of the current inference step, with fast convergence at simple steps and more extensive exploration at complex steps.

Furthermore, in order to ensure full exploration of the solution space, we first construct a density landscape based on the historical population, which shows the density distribution in the solution space. Then we consider not only the candidate steps' quality but also calculate their density by the density landscape. Though the non-dominated sort algorithm for multi-objective optimization in both quality and diversity dimensions, we select the Pareto-optimal reasoning steps to extend the search paths. The pseudo-code of the framework is given in Appendix E.2.2.

## 4.4 POPULATION CLUSTERING AND SELECTING

In the evolutionary process, to maintain population diversity while ensuring the preservation of high-quality solutions, we employ a clustering-based selection strategy. This strategy first clusters individuals in the population based on their answer labels, then selects the best individual from each cluster, ensuring that the final selected individuals are both high-quality and diverse.

This label-based clustering selection method ensures that we can prioritize the best solutions from each region of the solution space while maintaining population diversity. When the number of clusters is insufficient to meet the target selection count, the algorithm supplements by selecting the top highest-quality individuals from the remaining population, ensuring that the number of selected individuals reaches the expected target.

## 5 EXPERIMENTS

### 5.1 EXPERIMENTAL SETTINGS

**Implementation Details.**

We utilize Qwen2.5-7B-Instruct as the policy model and Qwen2.5-Math-PRM-72B Zhang et al. (2025b) as the process reward model (PRM), respectively. This reflects our philosophy of making smaller models more effective through collaboration rather than just scaling up. For all sampling processes, we maintain consistency with Qwen's PRM technical report Zhang et al. (2025b) by setting the temperature to 1.0 and top_p to 1.0. In all experiments, the sample number $k$ is set to 8. The quality threshold $\theta$ is set to 0.99. The maximum number of iterations during the evolution phase is set to 1. The bandwidth $h$ of Gaussian Kernel is set to 1, which indicates that in density calculations, only tokens from neighboring steps are computed.

**Evaluation Benchmarks.** To demonstrate the superiority of our designs in enhancing the mathematical reasoning capacity, we verify our method on several mathematical reasoning benchmarks, i.e., AIME-2024 MAA (2024), AIME-2025 MAA (2025), LiveMathBench Liu et al. (2024), MMLU-STEM Hendrycks et al., MATH-500 Lightman et al. (2023); Hendrycks et al. (2021), and GSM8K Cobbe et al. (2021). Note that we deliberately select a subset of the most difficult (level-5) problems from the MATH-500 dataset to rigorously assess our method when solving complex mathematical problems.

**Evaluation Metrics.** We employ the following metrics to evaluate the mathematical reasoning capabilities of the algorithms: (1) pass@k: the proportion of problems where at least one answer is correct among $k$ independent samples; (2) major@k: the proportion of problems where the most frequent answer among $k$ independent samples is correct; (3) prm@k: the accuracy of selecting the best answer from $k$ independent samples using the PRM; (4) sample@k: for step-level model generation, we sample $k$ candidate steps at each step and choice the step with highest score evaluated by PRM.

**Evaluation Details.** To evaluate the enhancement of SIER on LLM reasoning capabilities and its effectiveness in solution space exploration, we compare it against a set of strong baseline methods. Since our algorithm is based on swarm intelligence, which fundamentally differs from existing Multi-Agent Debate (MAD) frameworks. In addition, existing MAD methods perform worse than CoT methods Zhang et al. (2025a). In this case, we propose to perform comparisons with established CoT methods and Reward Guide Search (RGS) Zhang et al. (2025b), which improve the quality of each reasoning step through sampling. For CoT methods, prior approaches such as the Self-Consistency method employ a majority voting strategy, while the Best-of-N selection method utilizes a reward model to select the best one of the $N$ solutions that owns the highest score. Alternatively, we advocate

using the major@8 and prm@8 evaluation criteria as replacements for traditional majority voting and reward-based strategies. For the RGS method, we propose the use of the sample@8 evaluation criterion, which means that for each step, we sample 8 candidate solutions to consist with the sample number $k$.

## 5.2 PERFORMANCE COMPARISON

We provide a systematic comparison of our proposed SIER method with RGS and standard CoT approaches under various evaluation criteria in Table 1 . As shown in Table 1, our method consistently outperforms RGS and standard CoT approaches across various mathematical reasoning benchmarks, particularly on challenging datasets such as AIME and the level-5 subset of MATH-500. Specifically, SIER achieves the highest pass@8 and prm@8 scores across all benchmarks. For example, our method achieves a score of 26.7% on AIME-2024 and 30.0% under the pass@8 evaluation criterion, whereas the CoT method obtains 20.0% and 23.3% on the same benchmarks, respectively. This is because SIER enhances and preserves the diversity of solutions while maintaining the quality of understanding. Furthermore, even on the subset of MATH-500 with the level-5 difficulty, i.e., MATH-500 (level-5), our method also outperforms the CoT method. The main reason is that our method more thoroughly explores the solution space, obtaining higher-quality and more diverse solutions. Compared with the reward-based method, our method also performs better than RGS. This is because our density-assisted search explores a broader range of solution paths rather than focusing only on high-reward solutions, preventing the model from getting stuck in local optima.

We notice that our method consumes a relatively higher number of tokens compared to other methods, and the token usage increases with problem complexity. For example, our method consumes approximately 5.1k tokens on GSM8K, whereas other methods use around 3–4k tokens, which is relatively comparable. However, when applied to more complex datasets such as MATH-500, our method requires nearly five times as many tokens. This increased computational cost is inherent to SIER's methodology, which involves more extensive exploration of the solution space and additional refinement phases to improve the solutions' quality and diversity. It is encouraging that we can improve our reasoning ability by increasing the token data via extending the reasoning process, suggesting the potential to solve more complex reasoning problems.

Table 1: Performance Comparison. We present the performance of SIER, RGS, and CoT methods under various evaluation criteria. The best results are highlighted in bold. Tokens denote the average number of tokens consumed per task.

| Benchmark | SIER | | | RGS | | CoT | | | |
|---|---|---|---|---|---|---|---|---|---|
| | pass@8 | prm@8 | Tokens | sample@8 | Tokens | pass@8 | major@8 | prm@8 | Tokens |
| AIME-2024 | **26.7** | **23.3** | 40.8k | 20.0 | 12.4k | 20.0 | 16.7 | 16.7 | 8.61k |
| AIME-2025 | **30.0** | **13.3** | 35.0k | 10.0 | 11.9k | 23.3 | 10.0 | 10.0 | 8.42k |
| LiveMathBench | **60.7** | **47.9** | 45.6k | 47.1 | 8.15k | 55.0 | 39.3 | 46.4 | 7.28k |
| MMLU-STEM | **92.8** | **84.1** | 6.61k | 83.7 | 4.97k | 91.7 | 78.9 | 83.3 | 3.74k |
| MATH-500(level5) | **82.1** | **70.1** | 60.4k | 68.7 | 10.5k | 76.9 | 63.4 | 67.2 | 7.04k |
| MATH-500 | **93.0** | **86.2** | 32.3k | 84.6 | 6.43k | 89.8 | 81.8 | 84.2 | 5.40k |
| GSM8K | **97.4** | **95.8** | 5.10k | 95.3 | 3.98k | 97.0 | 93.3 | 95.3 | 3.13k |

## 5.3 ABLATION STUDIES

In this section, we conduct ablation studies to analyze the impact of key components in SIER. The key components of SIER lie in the candidate steps selection strategy with the guidance of fitness values and the density of the candidate steps, where the fitness values are obtained from the evaluator (PRM) and the density is calculated via the kernel density estimation (KDE) process. Besides, the evolutionary scheme in SIER plays a key role in further exploring the solution space and searching for high-quality and diverse reasoning paths.

In this case, we carefully elaborate three ablation variants, i.e., SIER w/o Fitness, SIER w/o Density, and SIER w/o Evolution, to independently examine the effectiveness of each component. To be specific, SIER w/o Fitness resorts to selecting candidate solutions solely based on the fitness score from the evaluator without the density information. Alternatively, SIER w/o Density adopts to select candidate solutions with the density information while ignoring the fitness score. SIER w/o Evolution represents a variant that removes the evolutionary mechanism entirely, making it essentially equivalent to standard CoT with PRM-based selection. This setup allows us to assess the critical contribution of the evolutionary search process within our framework.

Table 2: Ablation study. We validate the contributions of the fitness values provided by the evaluator (PRM), the density information estimated via kernel density estimation (KDE), and the evolutionary scheme in enhancing mathematical reasoning capability.

| Benchmark | SIER | | SIER w/o Fitness | | SIER w/o Density | | SIER w/o Evolution | |
|---|---|---|---|---|---|---|---|---|
| | pass@8 | prm@8 | pass@8 | prm@8 | pass@8 | prm@8 | pass@8 | prm@8 |
| AIME-2024 | **26.7** | **23.3** | 20.0 | 20.0 | 20.0 | 16.7 | 20.0 | 16.7 |
| AIME-2025 | **30.0** | **13.3** | 16.7 | 10.0 | 16.7 | 10.0 | 20.0 | 10.0 |
| LiveMathBench | **60.7** | **47.9** | 55.7 | 47.1 | 56.4 | 39.3 | 53.6 | 35.7 |
| MMLU-STEM | **92.8** | **84.1** | 88.6 | 84.0 | 89.1 | 76.1 | 88.6 | 80.7 |
| MATH-500(level5) | **82.1** | **70.1** | 77.6 | 69.4 | 79.1 | 67.2 | 76.9 | 67.2 |
| MATH-500 | **93.0** | **86.2** | 91.0 | 85.6 | 91.8 | 84.0 | 90.8 | 83.8 |
| GSM8K | **97.4** | **95.8** | 96.3 | 95.7 | 96.7 | 94.8 | 95.6 | 94.5 |

As shown in Table 2, the SIER method with full configurations performs the best across different ablation variants. In terms of the effectiveness of fitness values, it can help models avoid converging to local optima, increasing the solution diversities and consequently boosting the performance. Conversely, SIER w/o Density enhances diversity (sometimes achieving higher pass@8 than SIER w/o Fitness) but suffers from lower prm@8 scores due to a lack of explicit quality control. SIER w/o Fitness focuses on improving the quality of the solution (often obtaining higher prm@8 than SIER w/o Density), but has lower pass@8 scores due to too much focus on localized step scores of the solution at the expense of solution diversity.

Furthermore, the comparison with SIER w/o Evolution demonstrates the critical importance of the evolutionary mechanism in our approach. Without evolution, performance drops considerably across all benchmarks, especially on challenging tasks like AIME-2024 (pass@8: 26.7% vs. 20.0%; prm@8: 23.3% vs. 16.7%) and LiveMathBench (pass@8: 60.7% vs. 53.6%; prm@8: 47.9% vs. 35.7%). This substantial performance gap highlights the evolutionary search enhances the exploration of the solution space by further exploring unexplored regions through kernel density estimation, and maintains the quality of the solution localization step through PRM, enabling the discovery of higher-quality solutions that would otherwise remain inaccessible with standard sampling approaches. All these ablations together indicate the importance of maintaining high solution quality while encouraging sufficient diversity, effectively balancing exploration and exploitation to achieve superior overall performance.

## 5.4 Unsolved Problems Analysis

Our algorithm uses a threshold-based mechanism to control the quality of the solution. Specifically, the maximum number of iterations in the evolutionary phase is set to 1. This means that the task has been solved, and we will skip the evolutionary phase if the highest quality of the initial population exceeds the quality threshold $\theta$. Therefore, to further analyze the impact of the evolutionary phase, we focus on the unsolved problem identified by $\theta$ in depth in this section.

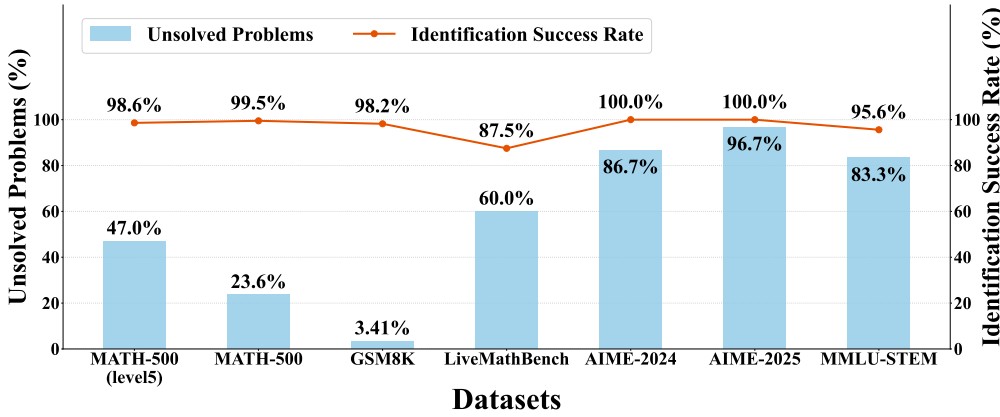

Figure 3: The percentage of unsolved problems identified by $\theta$ and the identification success rate across different datasets.

Fig. 3 presents the identification success rate of our algorithm in determining whether a problem has been solved. On AIME-2024 and AIME-2025 datasets, the algorithm achieves a 100% identification success rate. LiveMathBench shows a lower rate at 87.5%, possibly due to reward model limitations or dataset characteristics. Other datasets maintain high rates from 95.6% (MMLU-STEM) to 99.5% (MATH-500). The figure also shows the proportion of unsolved problems across datasets. Challenging datasets like AIME-2024 and AIME-2025 have higher percentages of unsolved problems, while MATH-500 and GSM8K show lower proportions, indicating our algorithm solves most tasks in these domains. This distribution reflects varying difficulty levels and suggests directions for future work.

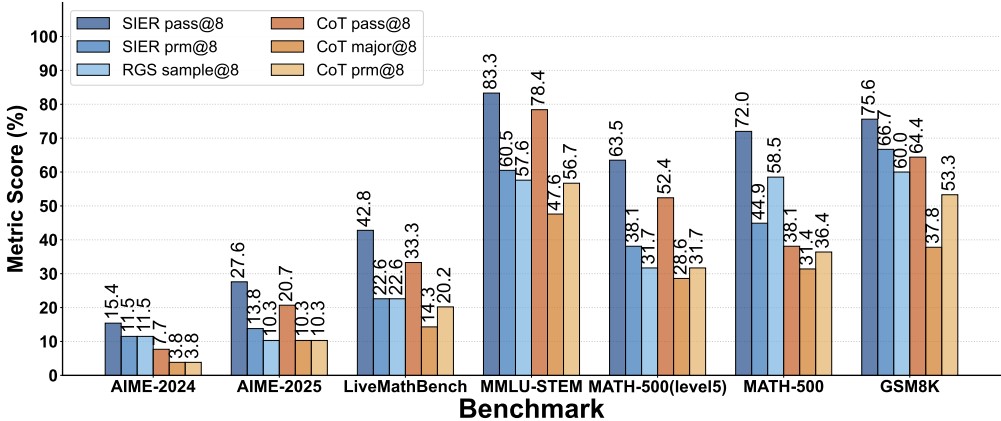

Figure 4: Performance comparison of SIER, RGS, and CoT variants on the unsolved problems.

Analyzing performance specifically on the subset of problems initially deemed unsolved (Fig. 4) provides further insights. SIER consistently achieves higher pass@8 scores than CoT across all unsolved datasets (e.g., AIME-2024: 15.4% vs. 7.69%; MATH-500(level5): 63.5% vs. 52.4%). SIER's prm@8 generally exceeds CoT's prm@8 (e.g., AIME-2024: 11.5% vs. 3.84%; MATH-500(level5): 38.1% vs. 31.7%). Compared to RGS's sample@8, SIER's prm@8 is competitive on some datasets (AIME-2024, LiveMathBench) but significantly higher on more challenging ones (MATH-500(level5): 38.1% vs. 31.7%; GSM8K: 66.7% vs. 60.0%). SIER also exhibits more stable performance across datasets compared to CoT and RGS. These results suggest SIER is more adept at discovering both correct (pass@8) and high-quality (prm@8) solutions even within challenging problem subsets. The dynamic balancing of solution diversity and quality, facilitated by density estimation and non-dominated sorting, likely contributes to its robustness and effectiveness on difficult tasks where other methods falter. SIER demonstrates a superior capability to find high-quality solutions in complex reasoning scenarios, particularly on difficult benchmarks where its advantage over CoT and RGS is most pronounced, highlighting the benefit of its integrated approach.

## 5.5 Parameter Analysis

The quality threshold $\theta$ serves as a crucial parameter in SIER that controls both solution acceptance criteria and guides the resampling strategy in its evolutionary mechanism. Experiments were conducted across multiple $\theta$ values ranging from 0.5 to 0.9 (with 0.99 as default) to evaluate its impact. Results demonstrated that higher $\theta$ values (0.9-0.99) achieve superior performance, especially on complex datasets, while lower values (0.5-0.8) maintain stable performance due to reduced activation of evolutionary mechanisms. (Details are given in Appendix G.1)

## 6 Conclusion

In this work, we propose a novel paradigm called Agent-based Swarm Intelligence (ASI) for enhanced reasoning, where LLM reasoning is conceptualized as a solution search process, and the swarm intelligence strategy is applied to find the global optima in the solution space. Accordingly, a density-driven framework is designed to employ kernel density estimation and non-dominated sorting for balancing solution quality and diversity when finding the optima. We conduct extensive experiments on seven challenging mathematical reasoning benchmarks, where our method consistently outperforms other methods under various evaluation criteria, showcasing the potential of swarm intelligence to enhance reasoning capabilities.

# 7 REPRODUCIBILITY STATEMENT

The algorithm is described in Section 4, with additional details in Appendix E. All experimental details, including parameters and implementation, are provided in Section 5.

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

## A   THE USE OF LARGE LANGUAGE MODELS (LLMS)

Large language models are used solely for language polishing. All innovative contributions presented are the independent work of the authors.

## B   SUPPLEMENTARY RELATED WORK

### B.1   MULTI-AGENT FRAMEWORKS

Large Language Model (LLM)-based multi-agent systems have gained significant attention for their ability to coordinate reasoning among multiple agents, enabling more structured, distributed, and interpretable problem-solving. A variety of frameworks have emerged with different strategies for agent collaboration and role differentiation.

CAMELLi et al. (2023) adopts a role-playing paradigm, where an "assistant agent" and a "user agent" follow distinct role prompts and engage in goal-directed dialogue. The system uses inception prompting to constrain conversational scope, facilitating self-consistent multi-turn collaboration without human intervention. However, the use of symmetric agents often leads to convergence on similar outputs, limiting diversity.

MetaGPTHong et al. simulates a software company by encoding Standard Operating Procedures (SOPs) into prompts. Agents assume specialized roles—such as product manager, architect, engineer, and QA—to generate modularized code through hierarchical task decomposition. This approach introduces modular thinking and reduces error propagation, but its applicability is mostly limited to software engineering tasks.

AutoGenWu et al. constructs a multi-agent execution graph with explicit message passing between agents. It supports automatic agent generation and flexible conversation flow design, enabling scalable orchestration and memory-aware communication. Nonetheless, the construction of effective message protocols often requires manual intervention, especially for complex tasks.

AgentVerseChen et al. offers a dynamic platform for instantiating heterogeneous agents with distinct capabilities and external tools. The system supports configurable agent types (e.g., searcher, planner, executor) and interaction patterns, allowing coordination in domains such as web navigation and embodied AI. Despite its flexibility, it demands significant task-specific engineering and infrastructure setup.

### B.2   SWARM INTELLIGENCE ALGORITHMS

#### B.2.1   NICHING METHODS

Evolutionary algorithms (EAs), such as genetic algorithm (GA) Wang et al. (2015); Cheng et al. (2018), differential evolution (DE) Lin et al. (2021); Jiang et al. (2023), and evolution strategy (ES) Luo et al. (2022); Xu et al. (2021), have become mainstream methods for solving multimodal (i.e., multi-peak) optimization problems (MMOPs) that have multiple global optima. Despite the success of traditional EA, when faced with complex, high-dimensional problems with multiple local optima (the number of local optima is much larger than the global optima), they usually converge only to the same optimal solution, which is most likely a local optimum. Therefore, assistant mechanisms are needed to facilitate the exploration of multiple promising regions in the search space.

Niching methods have become an important paradigm in MMOPs' research. These methods use different mechanisms to maintain population diversity and to identify multiple optima simultaneously. The theoretical foundation of niching methods resides in their capacity to partition the population into distinct sub-populations, each dedicated to the exploration of a specific region of interest. This partitioning can be achieved through various algorithmic approaches: crowding Mahfoud (1992), speciation Pétrowski (1996), and nearest-better clustering (NBC) Preuss (2010).

With these methods, the initialized population forms diverse subpopulations based on the relative spatial positions between individuals, distributing the solution space more adequately, however, there is still a risk of convergence to the same region during subsequent exploration of the evolution. With these methods, the initialized population forms diverse subpopulations based on the relative spatial

positions between individuals, distributing the solution space more adequately, however, there is still a risk of convergence to the same region during subsequent exploration of the evolution. Existing work has introduced more strategies to maintain the population's diversity to assist in searching during the exploratory phase of the population. Notable developments include penalty functions and exclusion mechanisms. Wei et al. (2022b) imposes penalties on sub-populations that fall into previously explored regions, while Ahrari et al. (2022) prevents sub-populations from conducting repetitive searches within the same area by defining a repelling radius for each sub-population.

Inspired by these algorithms, we introduce a density mechanism based on Agent-based Swarm Intelligence to generate high-quality and diverse solutions.

## C  ANALYSIS OF EXISTING WORK

### C.1  ANALYSIS OF MAD FRAMEWORK

Although the existing MAD framework is based on the hypothesis that LLMs can play different roles under various human prompts, unlike simple CoT, heterogeneous prompts can help agents collectively diverge their thinking and decide the final answer, which attempts to achieve two goals:

**Brainstorming:** Broaden the width of exploration of the solution space and obtain a variety of candidate solutions when facing complex problems.

**Selection:** Through interaction (e.g., rebuttal, evaluation, reflection), better solutions are finally selected.

However, existing evaluation work suggests that MAD frameworks do not perform as well as expected in terms of the above two goals. Zhang et al. (2025a) has experimentally found that many MAD frameworks perform worse than single-point methods and that self-consistency (i.e., by sampling the CoT results multiple times and voting for the best result) outperforms MAD frameworks significantly, as shown in Table 3.

Table 3: Performance results of different methods (Single-Agent (SA), Chain-of-Thought (CoT), Self-Consistency (SC) Wang et al., Society-of-Minds (SoM) Du et al. (2023), Multi-Persona (MP) Liang et al. (2024), Exchange-of-Thoughts (EoT) Yin et al. (2023), AgentVerse, and ChatEval Chan et al..) on GPT-4o-mini. Results higher than CoT for a given dataset are shown in red text, and results lower than CoT are in blue text.

| Dataset | SA(Single-Agent) | CoT | SC | SoM | MP | EoT | ChatEval | AgentVerse |
|---|---|---|---|---|---|---|---|---|
| MMLU | $65.33 \pm 0.93$ | $80.73 \pm 0.34$ | $82.13 \pm 0.66$ | $74.73 \pm 0.52$ | $75.47 \pm 0.84$ | $67.87 \pm 0.41$ | $79.13 \pm 0.90$ | $80.40 \pm 0.00$ |
| MMLU-Pro | $58.07 \pm 0.50$ | $62.80 \pm 0.99$ | $66.27 \pm 1.39$ | $62.80 \pm 1.02$ | $60.53 \pm 1.27$ | $61.20 \pm 0.65$ | $62.20 \pm 0.49$ | $62.07 \pm 0.52$ |
| CommonsenseQA | $79.47 \pm 0.25$ | $82.87 \pm 0.25$ | $83.80 \pm 0.28$ | $80.73 \pm 0.93$ | $68.07 \pm 1.57$ | $80.07 \pm 0.52$ | $81.07 \pm 0.84$ | $80.73 \pm 0.41$ |
| ARC-Challenge | $88.27 \pm 0.41$ | $93.53 \pm 0.41$ | $93.93 \pm 0.25$ | $90.80 \pm 0.43$ | $90.27 \pm 0.25$ | $86.40 \pm 0.28$ | $93.20 \pm 0.28$ | $92.47 \pm 0.09$ |
| AGIEval | $63.87 \pm 1.05$ | $66.40 \pm 1.30$ | $67.07 \pm 0.84$ | $64.33 \pm 0.34$ | $61.67 \pm 1.43$ | $65.07 \pm 0.66$ | $68.87 \pm 0.94$ | $63.87 \pm 1.23$ |
| GSM8K | $91.13 \pm 0.34$ | $93.60 \pm 0.82$ | $95.67 \pm 0.19$ | $94.93 \pm 0.34$ | $90.87 \pm 0.19$ | $91.40 \pm 0.57$ | $93.60 \pm 0.00$ | $92.73 \pm 0.50$ |
| MATH | $71.67 \pm 1.31$ | $72.87 \pm 1.20$ | $73.96 \pm 0.54$ | $75.40 \pm 0.71$ | $51.87 \pm 0.66$ | $75.93 \pm 1.23$ | $69.36 \pm 1.58$ | $64.49 \pm 1.38$ |
| HumanEval | $66.67 \pm 1.15$ | $78.05 \pm 1.49$ | – | $68.09 \pm 1.25$ | $63.01 \pm 2.30$ | $73.78 \pm 2.17$ | $71.75 \pm 0.76$ | $85.57 \pm 1.25$ |
| MBPP | $58.11 \pm 0.66$ | $62.26 \pm 0.84$ | – | $56.94 \pm 1.12$ | $45.78 \pm 0.80$ | $56.16 \pm 0.49$ | $53.70 \pm 0.55$ | $58.88 \pm 0.18$ |

Based on this research, we further analyze the brainstorming capabilities of the MAD framework when facing complex problems. We compare Society-of-Minds (SoM) Du et al. (2023), Multi-Persona (MP) Liang et al. (2024), Exchange-of-Thoughts (EoT) Yin et al. (2023), AgentVerse (Chen et al., 2024c), MA-ToT Haji et al. (2024), and CoT methods with the SIER proposed in this paper. "@8" represents that we sample 8 times for the generated results.

Specifically, we introduce the hit rate (HR) and diversity (Div.) metrics to evaluate this capability on the base model Qwen2.5-7 B-Instruct, and the temperature is set to 1.0. The hit rate metric represents the percentage of generated results that contain the correct answer, and the diversity metric represents the number of different results generated by the agents during the interaction process. The results are shown in Table 4:

CoT and SIER are evaluated with sample@8, while the other methods set the number of agents to 3 and the number of interaction rounds to 3. The experimental results show that the existing MAD frameworks not only perform poorly in terms of hit rate but also show limited diversity compared to the baseline CoT@8 method. For example, on the challenging AIME dataset, the diversity scores of

Table 4: Hit rate(HR) and Diversity (Div.) of the generated response by different methods across datasets.

| Method | AIME-2024 | | AIME-2025 | | MATH-500 | | GSM8K | |
|---|---|---|---|---|---|---|---|---|
| | HR(%) | Div. | HR(%) | Div. | HR(%) | Div. | HR(%) | Div. |
| CoT | 20.0 | 6.23 | 23.3 | 6.20 | 89.8 | 2.49 | 97.0 | 1.44 |
| SoM | 16.7 | 3.57 | 10.0 | 3.37 | 86.4 | 1.66 | 94.2 | 1.24 |
| MP | 3.33 | 2.10 | 3.33 | 2.10 | 77.6 | 1.42 | 90.8 | 1.19 |
| EoT | 13.3 | 4.20 | 6.67 | 2.03 | 61.6 | 1.79 | 95.1 | 1.32 |
| ToT | 16.7 | 2.97 | 6.67 | 3.37 | 79.4 | 2.14 | 94.6 | 1.42 |
| SIER | 26.6 | 7.00 | 30.0 | 7.27 | 93.0 | 2.67 | 97.4 | 1.46 |

CoT@8 are 6.23 and 6.20, respectively, while the diversity scores of the MAD frameworks (SoM, MP, EOT, and TOT) are significantly lower, ranging from 2.10 to 4.20.

The observations suggest a positive correlation between hit rate and diversity - the higher the diversity score, the higher the hit rate as well. For example, CoT@8 demonstrated higher diversity scores accompanied by higher hit rates compared to other MAD methods. Whereas methods with lower hit rates, such as MP (3.33% hit rate on both datasets), have much lower diversity scores (2.10). This correlation suggests that performance is closely related to the ability to generate diverse solutions when faced with complex problems that are difficult to solve directly.

Besides, to analyze the reasons for poor MAD performance, Wang et al. (2024) has identified two key problems that are relevant to the goal **Selection**: (1) **Judgment Error:** This occurs when the judger decides the final answer. If the answers vary between agents, the judger may choose the wrong option as the final verdict. This is especially true when the decision is made in a tie. (2) **Wrong Answer Propagation:** This error occurs when an agent, influenced by input from others, deviates from the correct answer and adopts an incorrect consensus, thus spreading the error further into the discussion. This is the most common mistake that can be made in a multi-agent discussion, even though they may have already gotten the correct answer.

Qwen's PRM report Zhang et al. (2025b) improves the performance of the CoT by using the PRM to choose the result with the highest score (prm@n) in place of traditional majority voting (major@n). Furthermore, they propose the Reward Guide Search (RGS) strategy. For each reasoning step of the COT, they sample multiple times and select the highest-score step by PRM. This approach mitigates the challenge in answer selection, but severely loses the diversity of generated results, and for complex problems, it is very easy to fall into the wrong local optimal solution and fail to search for the correct answer.

Therefore, we propose the SIER framework, which effectively accomplishes both of the above goals, increasing both the ability to brainstorm and the accuracy of the selection.

# D  THE DETAILS OF PRELIMINARY

## D.1  FAST NON-DOMINATED SORTING ALGORITHM

The pseudo-code of the Fast non-dominated sorting algorithm is shown in Alg. 1:

---

**Algorithm 1:** Fast Non-Dominated Sort

---

**Input:** A set of solutions $S = \{\mathbf{x}_1, \mathbf{x}_2, \ldots, \mathbf{x}_N\}$
**Output:** Non-dominated fronts $F = \{F_1, F_2, \ldots, F_k\}$

1 **for** $p = 1$ **to** $N$ **do**
2    $S_p \leftarrow \emptyset; n_p \leftarrow 0$ ; // Solutions dominated by $p$ and domination count
3    **for** $q = 1$ **to** $N$ **do**
4      **if** $\mathbf{x}_p$ *dominates* $\mathbf{x}_q$ **then**
5        $S_p \leftarrow S_p \cup \{q\}$
6      **else if** $\mathbf{x}_q$ *dominates* $\mathbf{x}_p$ **then**
7        $n_p \leftarrow n_p + 1$
8      **end**
9    **end**
10    **if** $n_p = 0$ **then**
11      $rank_p \leftarrow 1; F_1 \leftarrow F_1 \cup \{p\}$
12    **end**
13 **end**
14 $i \leftarrow 1$;
15 **while** $F_i \neq \emptyset$ **do**
16    $Q \leftarrow \emptyset$;
17    **foreach** $p \in F_i$ **do**
18      **foreach** $q \in S_p$ **do**
19        $n_q \leftarrow n_q - 1$;
20        **if** $n_q = 0$ **then**
21          $rank_q \leftarrow i + 1; Q \leftarrow Q \cup \{q\}$
22        **end**
23      **end**
24    **end**
25    $i \leftarrow i + 1; F_i \leftarrow Q$;
26 **end**
27 **Return** $F$;

---

# E  THE DETAILS OF METHODOLOGY

## E.1  AGENT-BASED SWARM INTELLIGENCE

Traditional swarm intelligence involves populations of individuals exploring solution spaces through fitness feedback and interaction. In the Agent-based Swarm Intelligence (ASI) framework, individuals are Large Language Model (LLM)-based agents. The generator $G$ (an LLM) produces solutions, and the evaluator $E$ (a Process Reward Model, PRM) assesses both overall solution quality and step-level reasoning. Agents leverage interaction information and evaluator feedback to effectively explore reasoning paths and navigate the solution space.

The core components of ASI include the generator $G$, evaluator $E$, and answer label extractor $T$, mathematically represented as:

$$x_g, c_g = G(q, p, M_g, t, s) \tag{3}$$

$$e_m, e_p = E(q, x_g, M_r, m) \tag{4}$$

$$x_{label} = T(x_g) \tag{5}$$

Where $q$ denotes the query, $p$ is the generated prefix, $M_g$ is the generative model, $t$ is the temperature parameter, $s$ represents stop words, $x_g$ is the generated text (solution), and $c_g$ is the token cost. $M_r$

is the process reward model, $m$ is the metric method, $e_m$ is the overall metric reward, $e_p$ is a list of step-level quality scores for steps in $x_g$, $T$ is the answer label extractor, and $x_{label}$ is the extracted answer label (e.g., from "\\boxed").

In the ASI framework, an individual $I_i$ and the population $\mathcal{P}$ are defined as:

$$I_i = \langle x_{g_i}, c_{g_i}, e_{m_i}, e_{p_i} \rangle \tag{6}$$

$$\mathcal{P} = \{I_1, I_2, \ldots, I_N\} \tag{7}$$

Where $\mathcal{P}$ is a population of $N$ individuals. Each individual $I_i$ contains the complete solution $x_{g_i}$, token cost $c_{g_i}$, solution quality score $e_{m_i}$, and step-level scores $e_{p_i}$.

Based on ASI, SIER generates candidate reasoning steps at the step level. It employs kernel density estimation to construct density landscapes and combines step-level evaluation with non-dominated sorting to select reasoning steps, guiding the population toward high-quality, diverse solutions.

### E.2 SIER FRAMEWORK

#### E.2.1 DENSITY CALCUALATION

Constructing the density landscape, i.e., calculating the current density value of each token using kernel density estimation, is the core of the SIER framework. The pseudocode is shown in Alg. 2.

---

**Algorithm 2:** Density Calculation

**Input:** History population $\mathcal{P}$, population size $N$, current step index $i_c$, bandwidth $h$, stop words $s$
**Output:** Token density dictionary $D$

```
1  H ← {};              // Initialize token history map H:  τ ↦ {step IDs}
2  foreach I ∈ P do
3  |   T ← ExtractTokens(I.x_g);      // Extract token sequence T from x_g
4  |   i ← 0;                         // Initialize step counter
5  |   foreach token τ ∈ T do
6  |   |   Add i to H[τ];            // Update step idx for token τ
7  |   |   if s ∈ τ then
8  |   |   |   i ← i + 1;         // Increment step if stop word s in token
9  |   |   end
10 |   end
11 end
12 D ← {};              // Initialize density map D:  τ ↦ density
13 foreach (τ, S) ∈ H do
14 |   if S = ∅ then
15 |   |   continue;
16 |   end
17 |   ρ_τ ← (∑_{s'∈S} K_h(i_c − s')) / N;  // Calculate density using step indices s' ∈ S
18 |   D[τ] ← ρ_τ;
19 end
20 return D;
```

$$\rho_\tau \leftarrow \frac{\sum_{s' \in S} K_h(i_c - s')}{N}$$

---

E.2.2   POPULATION EVOLUTION PHASE

In the population evolution phase, the pseudocode of the Density-assisted Search is shown in Alg. 3.

---

**Algorithm 3:** Density-Assisted Search

---

**Input:** Query $q$, generative model $M_g$, process reward model $M_r$, generator $G$, evaluator $E$,
metric $m$, quality threshold $\theta$, maximum iterations $\mathcal{I}_{\max}$, maximum steps $i_{\max}$, initial
population $\mathcal{P}_{\text{init}}$, temperature $t$, stop word $s$, end flag $f_{\text{end}}$, small batch size $b_s$, sample
size $k$, population size $N$, KDE bandwidth $h$

**Output:** Historical population $\mathcal{P}_{\text{hist}}$

1  $p_0 \leftarrow \emptyset; \mathcal{P} \leftarrow \{p_0\}; \mathcal{P}_{\text{hist}} \leftarrow \mathcal{P}_{\text{init}}$ ;          // Initialize paths and population

2  **for** $i = 1, 2, \ldots, \mathcal{I}_{\max}$ **do**

3      $i_s \leftarrow 0$ ;                              // Initialize reasoning step counter

4      **while** $i_s < i_{\max}$ **do**

5          $\mathcal{P}_{\text{new}} \leftarrow \emptyset$ ;                              // Store new active paths

6          **foreach** $u \in P$ **do**

7              **if** $f_{\text{end}} \in u[|u|]$ **then**

8                  **continue** ;                              // Skip completed reasoning path

9              $\{(n_1, c_1, e_{m,1}, e_{p,1}), \ldots, (n_{|u|}, c_{|u|}, e_{m,|u|}, e_{p,|u|})\} \leftarrow u$;

10             $x_{pre} \leftarrow \bigoplus_{l=1}^{|u|} n_l$;

11             $\mathcal{N} \leftarrow \emptyset; \mathcal{C} \leftarrow \emptyset; \mathcal{E}_m \leftarrow \emptyset; \mathcal{E}_p \leftarrow \emptyset; f_{\text{skip}} \leftarrow 0$;

12             **for** $j = 1$ **to** $k$ **do**

13                 $(n_{cand_j}, c_{cand_j}) \leftarrow G(q, x_{pre}, M_g, t, S)$ ;        // Generate candidate node

14                 $\mathcal{N} \leftarrow \mathcal{N} \cup \{n_{cand_j}\}; \mathcal{C} \leftarrow \mathcal{C} \cup \{c_{cand_j}\}$;

15                 $(e_{cand_m}, e_{cand_p}) \leftarrow E(q, x_{pre} \oplus n_{cand_j}, M_r)$;

16                 $\mathcal{E}_m \leftarrow \mathcal{E}_m \cup \{e_m\}; \mathcal{E}_p \leftarrow \mathcal{E}_p \cup \{e_p\}$;

17                 **if** $j \le b_s$ **and** $e_m \ge \theta$ **then**

18                     $u \leftarrow u \cup \{(n_j, c_j, e_m, e_p)\}; f_{\text{skip}} \leftarrow 1;$ **break**

19             **if** $f_{\text{skip}} = 1$ **then**

20                 **continue** ;                              // Skip if suitable node found

21             $D \leftarrow$ Alg. (2) $(\mathcal{P}_{\text{hist}}, N, i_s, h, s)$

22             $\mathcal{B} \leftarrow \{(-D[\mathcal{N}_l], \mathcal{E}_m[l]) \mid l = 1, 2, \ldots, |\mathcal{N}|\}$;

23             $F \leftarrow$ Alg. (1) $(\mathcal{B})$ ;                      // Non-dominated sort

24             **foreach** $l \in F_1$ **do**

25                 $p_{\text{new}} \leftarrow p \cup \{(\mathcal{N}[l], \mathcal{C}[l], \mathcal{E}_m[l], \mathcal{E}_r[l])\}; \mathcal{P}_{\text{new}} \leftarrow \mathcal{P}_{\text{new}} \cup \{p_{\text{new}}\}$;

26         $\mathcal{P} \leftarrow \mathcal{P}_{\text{new}}; i_s \leftarrow i_s + 1$;

27     **foreach** $u \in \mathcal{P}$ **do**

28         $\{(n_1, c_1, e_{m,1}, e_{p,1}), \ldots, (n_{|u|}, c_{|u|}, e_{m,|u|}, e_{p,|u|})\} \leftarrow u$;

29         $x_{\text{merged}} \leftarrow \bigoplus_{i=1}^{|u|} n_i; c_{\text{sum}} \leftarrow \sum_{i=1}^{|p|} c_i; e_{m_{total}} \leftarrow e_{p,|u|} \, e_{p_{total}} \leftarrow \bigoplus_{i=1}^{|u|} e_{p,i}$;

30         $I \leftarrow$ Eq. equation 6 $(x_{\text{merged}}, c_{\text{sum}}, e_{m,\text{avg}}, e_{r,\text{avg}}); \mathcal{P}_{\text{hist}} \leftarrow \mathcal{P}_{\text{hist}} \cup \{I\}$;

31 **return** $\mathcal{P}_{\text{hist}}$ ;                              // Return the final population

---

### E.2.3 Population Clustering Phase

The pseudocode of the Clustering is shown in Alg. 4.

---

**Algorithm 4:** Tag-based Clustering Selection Algorithm

---

**Input:** Current population $\Omega$, target selection count $k$
**Output:** Selected population $\Omega^*$

1   $\mathcal{C} \leftarrow$ Cluster individuals in $\Omega$ based on solution tags similarity;
2   $\Omega^* \leftarrow \emptyset$ ;                  // Initialize selection result set
3   $\mathcal{S} \leftarrow \emptyset$ ;           // Store clusters sorted by maximum fitness
4   **foreach** *cluster* $\gamma \in \mathcal{C}$ **do**
5     $\alpha \leftarrow$ Find individual with highest fitness in cluster $\gamma$;
6     $\phi \leftarrow$ Get fitness value of $\alpha$;
7     $\mathcal{S} \leftarrow \mathcal{S} \cup \{(\gamma, \alpha, \phi)\}$;
8   **end**
9   Sort $\mathcal{S}$ in descending order based on $\phi$;
10   $\lambda \leftarrow 0$ ;                    // Number of selected individuals
11   **foreach** $(\gamma, \alpha, \phi) \in \mathcal{S}$ **do**
12     $\Omega^* \leftarrow \Omega^* \cup \{\alpha\}$;
13     $\lambda \leftarrow \lambda + 1$;
14     **if** $\lambda \geq k$ **then**
15       **break** ;            // Target selection count reached
16     **end**
17   **end**
18   **if** $\lambda < k$ **then**
19     $\Omega^\dagger \leftarrow \Omega \setminus \Omega^*$ ;            // Unselected individuals
20     Sort $\Omega^\dagger$ in descending order based on fitness;
21     **while** $\lambda < k$ **and** $\Omega^\dagger \neq \emptyset$ **do**
22       $\beta \leftarrow$ Remove individual with highest fitness from $\Omega^\dagger$;
23       $\Omega^* \leftarrow \Omega^* \cup \{\beta\}$;
24       $\lambda \leftarrow \lambda + 1$;
25     **end**
26   **end**
27   **return** $\Omega^*$;

---

# F  THE DETAILS OF THE EXPERIMENT SETUP

## F.1  PROMPT

For all the algorithms mentioned in the experiments, we used the same prompt, which follows Qwen's official recommendations. The prompt is displayed as follows:

> **System Prompt:** Please reason step by step, and put your final answer within \\boxed{}.
> **User Prompt:** [Insert the question]

We differed from the traditional MAS approach in that we ditched the role-play method altogether and didn't use any fancy prompts. This enhances the robustness of the algorithm on different base models.

## F.2  MORE DETAILS OF REASON STEP BY STEP

Reasoning one step at a time and pausing at each reasoning step, by sampling multiple times and selecting high-quality and varied steps, is at the heart of the SIER framework. Specifically, we divide the inference steps by the stop word "$\backslash n \backslash n$", i.e., each time the generation of the model stops at "$\backslash n \backslash n$" as a single reasoning step, and splices this step with the previous result as a prefix for the next reasoning step generation.

Therefore, to implement our algorithm, it is required that the base model has the ability of text completion or prefix continuation, and the ability to set stop words. Fortunately, all open-source models and most of the closed-source models support the above abilities.

## G SUPPLEMENTARY EXPERIMENTS

### G.1 PARAMETER ANALYSIS

Our algorithm uses a threshold-based mechanism to control the quality of the solution. Specifically, the maximum number of iterations in the evolutionary phase is set to 1. This means that the task has been solved, and we will skip the evolutionary phase if the highest quality of the initial population exceeds the quality threshold $\theta$. In addition, in the population evolution phase, for the sampling of each inference step, we do not resample if the highest quality amount threshold of the candidate obtained from small batch sampling has exceeded the $\theta$. Thus, $\theta$ is the core parameter of SIER. In this section, we will further differentiate the value of the quality threshold $\theta$. In our experiments, the default value of $\theta$ is set to 0.99. We also tested values of 0.5, 0.6, 0.7, 0.8, and 0.9 to analyze their impact on performance. The results are shown in Fig. 5.

As observed in Figure 5, the quality threshold $\theta$ significantly impacts SIER's performance. Across most datasets, the model performs best when $\theta$ is set to higher values (0.9-0.99). This indicates that maintaining high-quality standards during candidate step selection is crucial for obtaining accurate solutions. Particularly on challenging datasets like AIME-2024, AIME-2025, and LiveMathBench, the setting of $\theta = 0.99$ notably outperforms lower thresholds.

Interestingly, we observe that when $\theta$ varies within the range of 0.5 to 0.8, the performance curves remain relatively stable on certain datasets. This stability can be attributed to lower thresholds reducing the likelihood of triggering the evolutionary phase of the algorithm. When $\theta$ is set too low, the probability that initial sampled steps have quality above the threshold increases substantially, significantly decreasing the probability of the resampling mechanism being triggered. This means that the evolutionary mechanism driving SIER's performance advantages is activated much less frequently. Consequently, the impact on overall results remains minimal within this parameter range. This finding validates the effectiveness of our algorithm's quality-diversity balance mechanism.

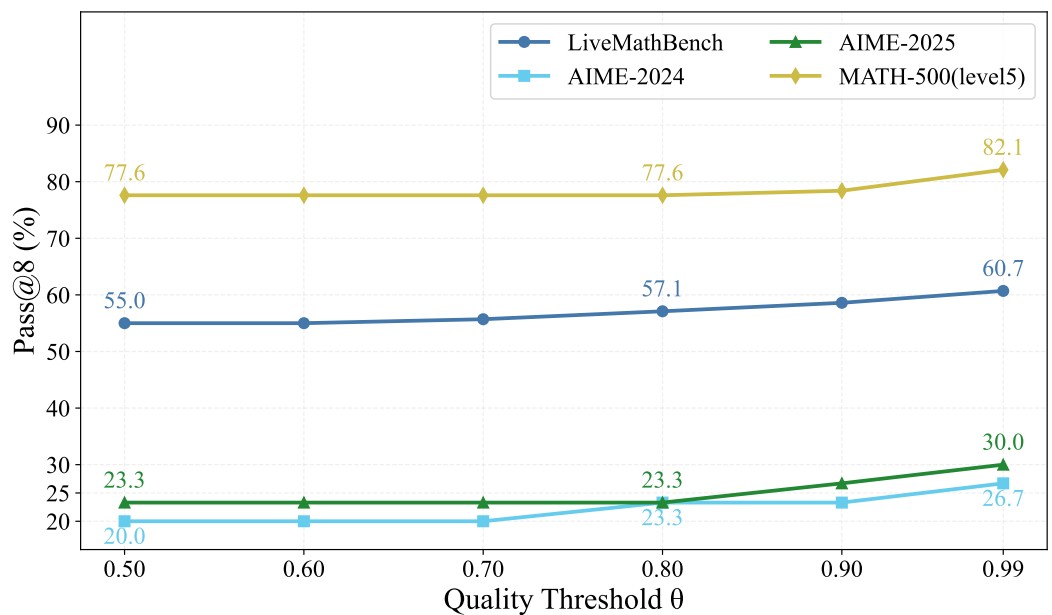

Figure 5: Impact of Different Quality Threshold $\theta$ on SIER's Performance (Log Scale).

### G.2 TREE SEARCH STRATEGY - REASONING VIA PLANNING

Reasoning via Planning (RAP) is a new reasoning framework for Large Language Models (LLMs) that aims to overcome the shortcomings of LLMs in generating task execution plans, and complex mathematical, logical, and common-sense reasoning.RAP does this by reorienting LLMs as world models and reasoning agents and combining them with planning algorithms based on Monte Carlo

Tree Search for reasoning in a wide space of strategic exploration in a wide range of reasoning spaces. During the reasoning process, LLM as an agent gradually builds a reasoning tree guided by LLM (as a world model) and task-specific rewards, and obtains highly rewarding reasoning paths by striking an appropriate balance between exploration and exploitation. We refer to the experimental setup in this paper and obtain the experimental results shown in Fig. 6.

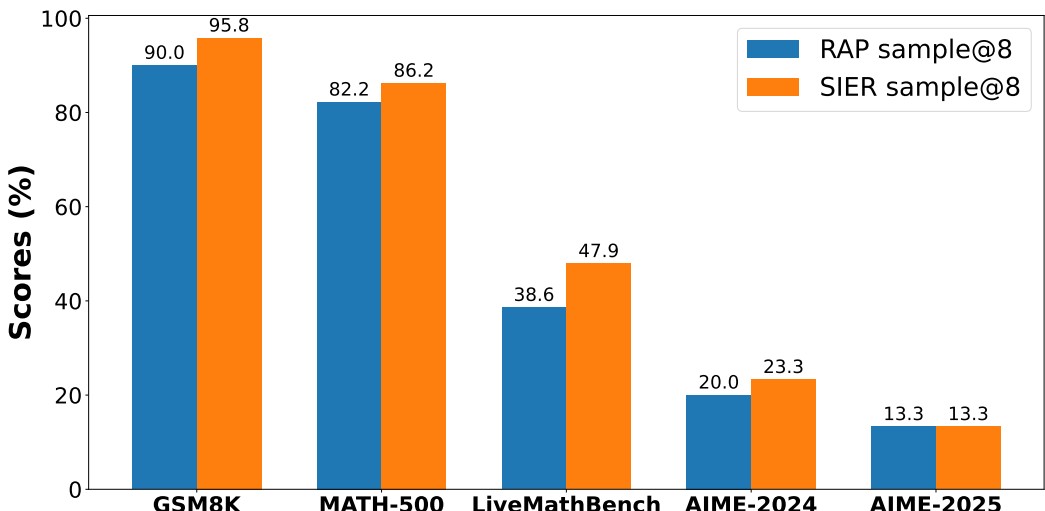

Figure 6: Performance comparison of SIER and RAP.

According to the experimental results, the SIER method proposed in this paper outperforms the RAP method on most datasets, especially on the GSM8K and LiveMathBench datasets. Specifically, the RAP method is inferior to SIER in terms of search efficiency and performance, probably due to the lack of reasonable sampling and pruning strategies. Moreover, RAP adjusts the values of the tree nodes through iterative optimization, and the model may concentrate on certain subtrees prematurely, i.e., it is easier to fall into local optima, which leads to difficulties in finding the global optima on complex problems. In contrast, the SIER method employs a more efficient search strategy and is able to explore the solution space better, resulting in higher performance.

## H  FURTHER DISCUSSION

We first consider an idealized scenario with a perfect evaluator providing accurate fitness scores reflecting true solution quality. The primary algorithmic challenge then becomes optimizing the search for high-fitness solutions. Effective algorithms can better identify global or near-global optima, making enhanced search efficiency crucial, especially for complex problems.

In practice, however, evaluators often function as black boxes. Influenced by training data and specific architectures (e.g., Process Reward Models (PRMs) or Outcome Reward Models (ORMs)), their assessments offer imperfect guidance. Fitness scores might not accurately represent solution potential, and reliance on them risks premature convergence. Different evaluator types present distinct challenges: PRMs, focused on process correctness, might undervalue high-potential solutions with unconventional steps, while ORMs, assessing only final outcomes, lack process insight, potentially rewarding flawed reasoning or penalizing sound but imperfect processes. Both limitations hinder reasoning capability development. These inherent constraints (PRM oversight, ORM detail limits) can trap algorithms in local optima reflecting evaluator biases, diminishing diversity, and impairing global exploration. Consequently, maintaining diversity while balancing exploitation/exploration is critical. Future algorithms must integrate diversity preservation mechanisms alongside fitness optimization to mitigate these challenges.

In summary, future research on agent-based swarm intelligence should prioritize two pivotal directions: enhancing the efficient search for high-quality, near-optimal solutions and devising robust strategies for diversity maintenance that counteract black-box evaluator limitations (e.g., PRMs, ORMs) and reduce local optima susceptibility. Progress in these complementary areas is essential for advancing LLM performance in complex reasoning tasks.

