# OpenReview forum: "Swarm Intelligence Enhanced Reasoning: A Density-Driven Framework for LLM-Based Multi-Agent Optimization"
_ICLR.cc/2026/Conference — Submitted to ICLR 2026_

### Official Review · Reviewer_k9Kr · 2025-10-17

**Soundness:** 2
**Presentation:** 3
**Contribution:** 2
**Rating:** 2
**Confidence:** 4

**Summary:**

This paper introduces SIER (Swarm Intelligence Enhancing Reasoning), a multi-agent framework for complex reasoning that draws inspiration from swarm intelligence. The method aims to overcome the limited solution diversity found in standard Chain-of-Thought (CoT) and Multi-Agent Debate (MAD) approaches. Its core mechanism is a density-driven search that uses Kernel Density Estimation (KDE) to avoid redundant exploration and a non-dominated sorting algorithm to simultaneously optimize for solution quality (evaluated by a Process Reward Model) and diversity (by favoring low-density paths). The framework is evaluated on several mathematical reasoning benchmarks, where it is primarily compared against CoT variants and a Reward Guided Search (RGS) baseline.

**Strengths:**

1. Novel Conceptual Framework: The paper's main contribution is the novel application of swarm intelligence principles to LLM-based reasoning. Framing the problem as a multi-objective optimization task that balances quality and diversity is a creative and theoretically interesting approach to address the well-known issue of cognitive homogeneity in multi-agent systems.
2. Sophisticated Technical Mechanism: The use of KDE to build a density landscape and non-dominated sorting to select Pareto-optimal reasoning steps is technically sound and non-trivial. This provides a principled mechanism for managing the exploration-exploitation trade-off at a granular, step-by-step level.
3. Solid Ablation Studies: The ablation experiments in Table 2 are well-designed and effectively demonstrate the necessity of both the quality (fitness) and diversity (density) components. The results clearly show that removing either one leads to a significant performance drop, validating the core design of the framework.

**Weaknesses:**

1. Comparison Against Simple or Outdated Baselines: The empirical validation primarily relies on comparisons against standard CoT prompting and a Reward Guided Search (RGS) strategy. These baselines are too simple and do not represent the current state-of-the-art in LLM reasoning. CoT is a general prompting technique, not a sophisticated search or agentic framework. While RGS is relevant, the paper fails to compare against more advanced agentic frameworks or models that have been fine-tuned with state-of-the-art RL or self-improvement techniques. This makes the reported performance gains appear larger than they might be against more competitive methods.
2. Trivial Gains on Practical Metrics for an Exorbitant Computational Cost: This is the most significant weakness of the paper.
  - The paper heavily relies on the pass@8 metric, which reflects the theoretical potential of the search space but not the final, usable output of the system. The more practical and meaningful metric is prm@8, which measures the accuracy of the single best answer selected by the reward model.
  - When we focus on prm@8, the performance improvement over the simple CoT baseline is marginal at best. For example, on the challenging MATH-500 (level 5) dataset, SIER (prm@8=70.1) provides a mere +2.9 percentage point improvement over the CoT baseline (prm@8=67.2).
  - This trivial gain comes at an exorbitant and impractical computational cost. On the same MATH-500 (level 5) dataset, the SIER framework consumes 60.4k tokens on average, which is nearly nine times the 7.04k tokens consumed by the CoT baseline. An almost 9x increase in compute for a less than 3% absolute gain in effective accuracy represents an extremely poor trade-off and calls into question the entire practical value of the proposed framework.

**Questions:**

1. Your baselines (CoT, RGS) are reasoning strategies, not state-of-the-art models or frameworks. Could you justify why more advanced multi-agent frameworks or models fine-tuned with recent RL techniques were not included for comparison?
2. Could you please address the cost-benefit analysis of your method? How can you justify the practical viability of a framework that, on MATH-500 (level 5), requires a nearly 9-fold increase in token consumption to achieve a marginal 2.9-point gain on the prm@8 metric, which reflects the actual output accuracy?
3. The pass@8 metric shows that your search finds correct answers, but the prm@8 metric shows your system often fails to select them. Does this not suggest that the primary bottleneck is the capability of the PRM, rather than the search strategy itself? How does your framework's performance change if a more powerful, or even oracle, evaluator is used?

---

### Official Review · Reviewer_RBBn · 2025-10-24

**Soundness:** 2
**Presentation:** 2
**Contribution:** 3
**Rating:** 2
**Confidence:** 3

**Summary:**

CoT and MAD methods often fail for complex tasks because of low solution diversity and convergence to local optima. The key idea of this manuscript is to treat the LLM reasoning as an optimization problem over the solution space and apply swarm intelligence methods to avoid local minima. They implement this in a framework SIER, which uses, among other steps, KDE to construct a density landscape, guiding agents towards low-density (=underexplored) solution regions. They evaluate their framework on several math reasoning benchmarks such as AIME-2024, comparing against e.g. COT and RGS, which they outperform. As a trade-off, their approach has a multiple of the token usage.

**Strengths:**

The idea seems novel: reframe LLM reasoning as a swarm intelligence-type optimization problem, then use the methods available for that kind of problem. Kernel densitry estimation is a powerful way to balance exploration vs exploitation in a clear manner, which is a major weakness for other more experimental approaches. Their methodology is well-edfined and powerful, with a lot of mathematical grounding. They also used a good number of reasoning benchmarks to evaluate on, including ones that are comparatively easier (GSM8K) and harder (MATH-500). Their Pareto front selection is a good (interpretable) approach for reasoning quality. Ranked the contribution as good because the idea of swarm intelligence in this context seems to have a lot of promise, even if I don't see it overall in this version.

**Weaknesses:**

The most significant drawback is the computational inefficiency. I 5x token cost on complex datasets is, unfortunately, outweighing the contribution this paper would otherwise be. Scalability and practical deployment cost is just not feasible with such a 5x factor, or at least, it would need to be more strongly argued for.

Likewise, it would have to be shown whether such a factor is limited to math reasoning, and how well (and with what inefficiency factor) the framework works on more general scientific or multimodal reasoning tasks.

The components of the framework are not really novel. This is not truly a drawback, but somewhat diminishes the overall impact. This is a novel combination of tried-and-tested methods (KDE, pareto front, evolutionary selection), which worked to a degree, but with a in my opinion prohibitive cost factor.

I think the quantitative metrics chosen are maybe a bit superficial (pass, prm), but I would like to have seen morme qualitative analyses on the reasoning paths, and the types of errors that are corrected.

**Questions:**

SIER uses far more tokens (up to 5× CoT). Can the authors explain where this overhead comes from?

Kernel density estimation is O(N^2) How is this computationally feasible for larger problems?

Have the authors compared wall-clock time or FLOPs, not just token counts, against CoT and RGS?

If all methods were constrained to the same token budget, would SIER still outperform?, and related:
The accuracy gains (˜3–5%) seem modest relative to the increased computation. Can the authors quantify efficiency in terms of improvement per 1 k tokens?

---

### Official Review · Reviewer_HocR · 2025-10-31

**Soundness:** 2
**Presentation:** 2
**Contribution:** 2
**Rating:** 4
**Confidence:** 3

**Summary:**

This paper proposes a novel framework, Swarm Intelligence Enhancing Reasoning (SIER), which integrates swarm intelligence into the reasoning process of Large Language Models (LLMs) to enhance their problem-solving capabilities, particularly for complex reasoning tasks. The authors address the limitations of traditional methods like Chain-of-Thought (CoT) and Multi-Agent Debate (MAD), which struggle to find optimal solutions due to lack of diversity and vulnerability to local optima. The SIER framework utilizes an Agent-based Swarm Intelligence (ASI) approach, where multiple agents explore a solution space collaboratively. The framework enhances the exploration by using kernel density estimation and non-dominated sorting to balance solution quality and diversity. Through extensive experiments on mathematical reasoning benchmarks, SIER consistently outperforms CoT and reward-guided approaches, especially on complex problems.

**Strengths:**

1. The use of kernel density estimation and non-dominated sorting ensures that the exploration of the solution space is both diverse and of high quality, avoiding the pitfalls of convergence to local optima.
2. The framework is extensively tested on challenging benchmarks like AIME, MATH-500, and GSM8K, with significant improvements over traditional methods, particularly for more difficult problems.
3. The dynamic control of the exploration process through quality thresholds and flexible termination criteria makes the framework adaptable to different problem complexities and computational resources.

**Weaknesses:**

1. The framework requires higher computational resources, especially when dealing with more complex problems (e.g., MATH-500), as it involves more extensive exploration of the solution space. The increased token usage could be a limitation for large-scale applications.
2. The effectiveness of the framework relies heavily on the quality of the Process Reward Model (PRM). If the evaluator is biased or inaccurate, it may still lead to suboptimal solutions, especially in cases where the PRM is unable to accurately assess the reasoning steps.
3. The paper focuses primarily on enhancing LLMs with swarm intelligence but does not explore how other optimization techniques could be integrated or compared within this framework.
4. Although the framework includes mechanisms to avoid getting trapped in local optima, there remains the inherent risk that the swarm intelligence and the evaluator could lead the process in directions that do not yield the optimal solution, particularly in the absence of perfect evaluators.
5. The computational cost associated with the Swarm Intelligence Enhancing Reasoning (SIER) framework is notably high, particularly due to the complexity of the kernel density estimation (KDE) and non-dominated sorting used to maintain diversity and evaluate solutions . Although this is addressed with a dynamic control mechanism, the solution still lacks clear suggestions for making the algorithm more computationally efficient.

**Questions:**

1. The effectiveness of the framework seems to rely heavily on the quality of the Process Reward Model (PRM). Could you clarify how robust the framework is when the PRM is inaccurate or biased? What strategies can be employed to reduce the risk of suboptimal solutions when the evaluator fails to provide accurate feedback on reasoning steps?
2.The paper primarily focuses on enhancing LLMs with swarm intelligence but does not explore how other optimization techniques (e.g., genetic algorithms, simulated annealing) could be integrated or compared within this framework. Could you discuss the potential for incorporating alternative optimization methods, and how these might complement or enhance the current approach?
3.While the framework includes mechanisms to avoid local optima, there is still the inherent risk that the swarm intelligence and evaluator could lead the process in directions that do not yield the optimal solution. Could the authors provide more detailed discussions on how the algorithm might behave in particularly challenging or ambiguous problem scenarios where even the enhanced diversity mechanisms fail to produce optimal solutions?
4.The computational cost of the Swarm Intelligence Enhancing Reasoning (SIER) framework is noted to be high, particularly due to the complexity of kernel density estimation (KDE) and non-dominated sorting. Although the paper mentions a dynamic control mechanism, there is no clear suggestion for improving computational efficiency. Could the authors propose or experiment with specific techniques for reducing the computational load, especially in large-scale or high-dimensional problems?

---

### Official Review · Reviewer_CVex · 2025-11-01

**Soundness:** 1
**Presentation:** 3
**Contribution:** 3
**Rating:** 2
**Confidence:** 4

**Summary:**

SIER presents a novel alternative to CoT and MAD methods. SIER uses a lighter policy model guided by a heavier evaluator model to generate answers through a diversified search space. Authors claim that SIER performs better than both CoT and MAD, while using up to 5x more tokens.

**Strengths:**

The paper demonstrates sound methodology through its novel SIER approach, which advances test-time scaling by introducing sample diversity mechanisms and multi-dimensional evaluation criteria beyond traditional methods like MAD and CoT. The authors provide convincing initial evidence of SIER's superiority in mathematical reasoning tasks, supported by a clearly articulated algorithm that details how diverse sampling and comprehensive evaluation work together to improve reasoning outputs. While the current results are promising and the theoretical framework is well-constructed, the soundness of the contribution would be strengthened by demonstrating that SIER's advantages persist across different model architectures and extend beyond mathematical reasoning to other problem domains—validation that would confirm whether this represents a broadly applicable advancement in test-time scaling rather than a domain-specific optimization.

**Weaknesses:**

I believe that SIER method has a strong potential, and it was compelling to see that SIER had superior performance across several mathematical reasoning benchmarks. But there is not enough evidence to make strong claims yet. The only policy-reward model combination evaluated was Qwen2.5-7B-instruct with Qwen2.5-Math-PRM-72B. It's also unclear unclear what models were used for the RGS and CoT methods used to compare against SIER (Table 1). From the wording of the paper I'm assuming it was done with Qwen2.5-7B-instruct, but this feels unfair since SIER is "boosted" by the intelligence of Qwen2.5-Math-PRM-72B so we should have CoT on Qwen2.5-Math-PRM-72B for comparison as well. I'd also like to know how the SIER method fare as we scale up model parameter size, or use other frontier LLMs.

**Questions:**

1. Exactly what were the objectives used to judge for Pareto-optimal solutions?
2. KDE was used here to diversify the responses - how does this compare with simply increasing the policy temperature?
3. How closely does the strength of the PRM affect the quality of SIER responses?

Grammar issue: line 318-319

---

### Meta-Review · Area_Chair_DTEU · 2025-12-28

**Summary:**

The SIER framework is conceptually novel, introducing swarm intelligence to LLM reasoning and improving solution diversity. Authors did not response to reviews so this paper should be rejected.

**Reviewer Concerns:**

While it outperforms standard methods like CoT and RGS on math benchmarks, the computational cost is extremely high for only marginal accuracy gains (2–3%), and stronger baseline comparisons are lacking. The approach also relies heavily on evaluator quality. Overall, the practical utility is limited; I recommend rejection. Further efficiency improvements, stronger baselines, and broader task evaluation are needed.

**Reviewer Scores:**

Reviewers provided clear evidence for their insightful concerns and authors did not response.

---

### Decision · Program_Chairs · 2026-01-26

Reject